# Nano-Biomaterials for Retinal Regeneration

**DOI:** 10.3390/nano11081880

**Published:** 2021-07-22

**Authors:** Rahul Sharma, Deepti Sharma, Linda D. Hazlett, Nikhlesh K. Singh

**Affiliations:** 1Integrative Biosciences Center, Wayne State University, Detroit, MI 48202, USA; rahuldmt@gmail.com (R.S.); dsharma25@wayne.edu (D.S.); 2Department of Ophthalmology, Visual and Anatomical Sciences, Wayne State University School of Medicine, Detroit, MI 48202, USA; lhazlett@med.wayne.edu

**Keywords:** nanoparticles, nanodisks, scaffolds, nano-biomaterials and retina, nanoscaffolds and retinal regeneration, nanoparticles and retinal regeneration

## Abstract

Nanoscience and nanotechnology have revolutionized key areas of environmental sciences, including biological and physical sciences. Nanoscience is useful in interconnecting these sciences to find new hybrid avenues targeted at improving daily life. Pharmaceuticals, regenerative medicine, and stem cell research are among the prominent segments of biological sciences that will be improved by nanostructure innovations. The present review was written to present a comprehensive insight into various emerging nanomaterials, such as nanoparticles, nanowires, hybrid nanostructures, and nanoscaffolds, that have been useful in mice for ocular tissue engineering and regeneration. Furthermore, the current status, future perspectives, and challenges of nanotechnology in tracking cells or nanostructures in the eye and their use in modified regenerative ophthalmology mechanisms have also been proposed and discussed in detail. In the present review, various research findings on the use of nano-biomaterials in retinal regeneration and retinal remediation are presented, and these findings might be useful for future clinical applications.

## 1. Introduction

Nanotechnology is one of the emerging tools used to design regenerative medicine approaches that can target specific organs or cells in humans. In this context, the first technique of tissue regeneration was carried out in 1902 by Alexis Carrel [1], in which he replaced or repaired inflamed tissues by transferring cells and tissue constructs to the body [2,3,4]. In the last decade, we have observed an omnidirectional use of nanomaterials in tissue regeneration due to their reduced, nanoscale morphology. The physical, chemical, optical, and electronic properties of nanostructures are dependent on their morphological (shape and size) and compositional attributes. Furthermore, we can manipulate and regulate these properties not only for the treatment and diagnosis of disease but also for the monitoring and control of the disease [5,6,7,8].

The eye is a unique sensory organ that anatomically and physiologically has a direct connection to the sensory nervous system in the brain. The eye is a two-piece unit composed of an anterior and a posterior segment. The anterior ocular segment is made up of the cornea and the lens, and the posterior segment comprises the retina, choroid, and optic nerve. Drug administration to both segments of the eye is accomplished through different routes, and effective delivery is highly challenging for modern ophthalmology. Like the brain, the posterior segment of the eye has several anatomic and dynamic protective barriers, made up of the tight junctions present in the ciliary body epithelium, endothelial cells of the iris, retinal pigment epithelium, and the retina (Figure 1). In addition to providing resistance to several toxins, harmful substances, and microorganisms, these barriers also preclude the diffusion of drug particles to the targeted ocular segments [9].

Retinal degeneration is a clinical manifestation of various retinal degenerative diseases, which include age-related macular degeneration (AMD), retinitis pigmentosa, Stargardt’s disease, and diabetic retinopathy [10]. The degeneration of RPE and photoreceptors are the most common outcomes of these diseases, which further leads to a visual disability or vision loss. The existing therapies for these retinal degenerative diseases are only able to delay the progression of these diseases. The recent advances in the delivery of syngeneic or allogeneic tissues/cells have become the choice of therapy to regenerate retinal tissue [11]. Efforts are underway to use engineering approaches to regenerate retinas that have been lost or damaged due to various degenerative diseases. To overcome the limitations of conventional eye drops and of intraocular invasive injections, several ophthalmic formulations have been proposed, such as drug-loaded nanoparticles/nanocarriers. Nanoparticles, which are submicron-sized particles ranging from 10 to 1000 nm, can provide a versatile platform for drug delivery. Drugs can be loaded into such nanoparticles by attachment to the matrix, or the drug can be dissolved, encapsulated, or entrapped within their nanomorphologies. In various stages of clinical studies, the Food and Drug Administration (FDA) has approved nearly 250 nanomaterial-based medical products [12]. With recent advancements, nanomedicine approaches to the regeneration of tissues have been particularly focused on using certified functional nanomaterials. These engineered nanomaterials not only deliver cells and tissues but also monitor tissue regeneration processes in real time, thereby improving the overall therapeutic efficiency. The compatibility of biological organs with various nanomaterials, such as nanoparticles (NPs), nanowires (NWs), and hybrid nanostructures, has enhanced the probability of their use in biomedical applications, especially in retinal regeneration (Figure 2) [13,14,15,16,17,18]. Among these, nanoparticles such as gold NPs (AuNPs) and magnetic iron oxide nanoparticles (MIONPs) are widely used in preclinical and clinical settings due to their well-established imaging and therapeutic properties [19,20]. Furthermore, because of their physical and chemical properties, nanoparticles have recently been introduced as contrast enhancement agents for many imaging modalities such as MRI [21,22,23,24,25], fluorescence imaging [26], photoacoustic imaging [27], ultrasound imaging, and computed tomography (CT) [28,29,30,31,32,33,34,35]. In recent years, modified nanoparticles have been in high demand for their use in clinical practices for in vitro metabolic assays. In this context, studies have shown that gold nanoparticles deposited on the plasmonic chip and a porous silica-based plasmonic nanoreactor are useful for the metabolic analysis of biofluids [36,37]. Some studies have used nano-biomaterials to treat antibiotic-resistant bacterial infections [38]. Furthermore, the use of platinum nanoreactor, polymer@Ag-assisted, and bimetallic alloy-based laser desorption/ionization mass spectrometry showed its usefulness for metabolic fingerprinting and disease diagnosis [39,40,41].

In this review, we not only summarize the regenerative approaches used for retinopathy with the advancement of nanoscience and nanotechnology, but also shed some light on the future challenges and perspectives in retinal regeneration.

## 2. Nanomaterials for Retinal Regeneration

Nanomaterials are useful for various retinal applications, where their functional significance relies on their material properties. In the present section, we will discuss the importance of nanoparticles, nanowires, and hybrid nanostructures in retinal regeneration, summarized in Table 1.

**Table 1 nanomaterials-11-01880-t001:** Details of various nanostructures and their morphologies for targeting specific tissues or cells for retinal regeneration.

Nanostructure	Nanomaterial	Size Range (nm)	Target Tissue/Cells	Ref.
Nanoparticles	Gold (Au) (diameter)	3–5	Choroidal and retinal endothelial cells	[42]
		10–12	Retina of rabbit	[43]
		10–20	Photoreceptor precursor transplantation	[44]
		80	Retinal cells	[45]
		20–80	Nucleus and mitochondria of retinal cells	[46]
		5–20	Blood–retinal barrier	[44,47,48]
	Gold (Au) nanodisk	Thickness: 20 Diameter: 160	Retina	[47]
	Silver (Ag) (diameter)	20–80	Bovine retinal endothelial cells	[49]
		40–50	Porcine retinal endothelial cells	[50]
	Superparamagnetic iron oxide nanoparticles	Diameter: 5–20	Retina	[51]
	Magnetite	10	Retina and cells	[52,53]
NWs	Poly (ε-caprolactone) (PCL) membranes	Length: 2500	Implantation into subretinal space	[54]
	Gallium phosphide (GaP)	Length: 500–4000	Retinal cells	[55]
	*n*-type silicon	Length: 4400	Retinal cells	[56]
	Gold (Au) nanorods	Thickness: 10–35	Retinal cells and photoreceptors	[57]
Hybrid nanostructure	Gold NPs coated over titania (TiO_2_) NWs	Au NP diameter: 5–15 TiO_2_ NW length: 2000	Artificial photoreceptors	[54,58,59,60,61]
	Gallium phosphide (GaP) rod and cone	Length: 20–2500	Ganglion cells, and bipolar cells	[55]
	Gold NPs coated over silicon NWs	Au NP diameter: 5–10 NW length: 500–2500	Artificial photoreceptors	[62,63]
	Thin film functionalized with the NPs	Diameter: 5–50	Photoreceptors	[64,65]
	*p*–*n* junction silicon NWs	NW length: 10–120	Membranes of live bipolar cells	[66]
	Au-coated carbon nanotube (Au-CNT)	Au NP diameter: 5–20 CNT length: 500–2500	Subretinal space of mice	[67]
	Iridium oxide (IrOx) combined with reduced graphene oxide	IrOx diameter: 2–25 CNT length: 2–2500	Subretinal implant into live mice	[68]
	Iridium oxide (IrOx) coated with CNT	IrOx diameter: 5–25 CNT length: 500–2500	Retinal cells/tissues	[52,69,70,71,72]
	Core–shell-structured β-NaYF4:20%Yb, 2%Er@β-NaYF4 nanoparticles	Diameter: 30–40	Subretinal space of mice	[15]
Nanoscaffolds	Natural polymer: gelatin, fibrin, chitosan, laminin, and hyaluronic acid	Diameter/porosity: 100–200	Extracellular matrix and cell attachment	[73,74,75,76,77,78]
	Synthetic polymer: poly (lactic-*co*-glycolic acid) (PLGA), poly (ε-caprolactone) (PCL), poly (L-lactic acid) (PLA), polyimide, and poly (l-lactide-co-ε-caprolactone)	Diameter/porosity: 50–500	RPE, biological activity, extracellular matrix, and cell attachment	[79,80,81,82]
	Biohybrid: nanofibers of Bruch’s membrane	Diameter/porosity: 100–200	RPE and biological activity	[83]

### 2.1. Nanoparticles

Nanoparticle-based gene and drug delivery to retinal cells has been harnessed to treat various eye diseases [44,84,85,86,87,88,89,90,91]. The various transport mechanisms that nanoparticles employ to cross the blood–retinal barrier are shown in Figure 3. Nanoparticles absorb or scatter light at specific frequencies/wavelengths as a function of their physical and chemical characteristics. These properties of nanoparticles are well suited for bioimaging and to treat cancer by using near-infrared-triggered photothermic therapy (PTT) [92]. Due to the low absorption coefficients of hemoglobin and water, the penetration of near-infrared (650–900 nm) rays in tissues is very high, allowing the use of near-infrared rays for nanoparticle stimulation without damaging the tissue [93]. Gold-nanoparticle-based intravitreal injection is used for retinal imaging and for the inhibition of retinal neovascularization to treat macular degeneration [89,90,91]. Gold nanoparticles have been shown to be promising carriers of anti-vascular endothelial growth factor (VEGF) antibodies to targeted sites in the eye [44,90,91]. It has also been shown that gold nanodisks attenuate pathological retinal angiogenesis in a murine model of oxygen-induced retinopathy [47]. In other words, gold nanodisks are more optically suitable due to their capability to produce signals that are detectable irrespective of the direction of polarization of the light source. Furthermore, gold nanodisks have been shown to minimize reactive oxygen species (ROS) production compared to gold nanoparticles or gold nanorods. In addition, silver nanoparticles have also been shown to reduce VEGF-induced cell migration and proliferation of bovine retinal endothelial cells. Furthermore, in vivo use of silver nanoparticles significantly reduced retinal neovascularization and angiogenesis within metastasizing cells [58].

### 2.2. Nanowires

Engineered nanostructural materials are essential for the development of advanced retinal applications. Among them, nanowires have been reported for retinal applications in recent years [59]. It has also been demonstrated that the structure and morphology of nanowires are similar to those of photoreceptors, and the photoabsorption and charge separation properties of nanowires are comparable to those of photodetectors or solar cells [60]. The gold-nanoparticle-decorated titania (Au-TiO_2_) nanowire acts as an artificial photoreceptor that restores the light responses in a photoreceptor-degenerated retina. The use of nanowires with poly (ε-caprolactone) (PCL) scaffolds for the delivery of retinal progenitor cells resulted in increased differentiation and migration of these cells into both degenerated and normal retinas [54,61]. Nanowires made of gallium phosphide have been shown to support the long-term survival of photoreceptors (rods and cones), ganglion cells, and bipolar cells [55]. Furthermore, nanowires coated with poly-L-ornithine displayed a significant difference in cell morphology and cell adhesion compared to flat substrates [64,94,95]. Silicon nanowires (Si NWs) coated with gold nanoparticles were more effective in photoreceptor stimulation, with little or no damage to the cells/tissues [56,62]. Another type of nanostructure is thin films, and the photoelectric signals initiated by these thin films mimic the response of the retina to visible light [55,63,96,97].

Nanowires not only sense the incoming light but also transfer electrical signals within rod and cone cells [65]. To create nanowires that can transfer electrical signals, researchers have used *n*-type and *p*-type silicon materials. These silicon materials sense light and transform it into electrical signals [14,98]. Furthermore, the one-dimensional morphology of silicon nanowires is better suited to sense light and convey the signals to the various retinal layers to correct the visual impairment [14]. The incorporation of *n*-type and *p*-type silicon in nanowires has made it possible for the nanowires to convert light signals to electrical signals and then transfer them into the membranes of live bipolar cells for vision recovery [99].

### 2.3. Hybrid Nanostructures

A vast range of nanomaterials and nanostructures have been explored as neural interfaces in retinal physiology; still, no single material has been successful in mimicking the biological, mechanical, and electrical properties of the retina. In recent years, many hybrid approaches have been designed to explore the merits of many materials while at the same time suppressing their demerits. Recently, Tang et al. demonstrated that artificial photoreceptors made of gold-nanoparticle-decorated titania (Au–TiO_2_) nanowire arrays were able to absorb light, generate photovoltage, and process visual information in a photoreceptor-degenerated retina [14]. Not only is nanowire arrays’ rough morphology useful for their association with cultured neurons, but they are also biocompatible or (photo)chemically stable for over 2 months when used as a subretinal implant in mice [66]. The use of a gold coating on carbon nanotubes (Au-CNTs) further enhanced their surface area and electrical and mechanical adhesion [100]. Iridium oxide–carbon nanotube hybrids (IrOx-CNT) were reported to have a high effective surface area and much higher charge storage capacities compared to pure iridium oxide [67]. Furthermore, the hybrid coatings formed by combining iridium oxide with reduced graphene oxide or graphene oxide exhibited 10% higher charge storage capacities than those of pure iridium oxide and iridium oxide–carbon nanotube hybrids, indicating superior electrochemical stability [52,68,69,70,71]. 

Hybridization of conducting polymers with carbon nanotubes has also provided a highly biocompatible, electrically active, and mechanically strong coating. An electrochemical study showed that a poly (3,4-ethylene dioxythiophene)–multiwalled carbon nanotube hybrid coating provides superior stability compared to a pure polystyrene sulfonate coating, with only 2% loss of charge storage capacity [71]. Grooved single-walled carbon nanotube/poly (3,4-ethylene dioxythiophene) composite films not only showed increased biocompatibility but also provided physical cues for neuronal cell proliferation and differentiation [72]. The inclusion of hydrogels and biomolecules to the hybrid coating further enhanced the number of active channels in an in vivo recording compared to a conventional coating [52]. Furthermore, the use of anti-inflammatory agents with hybrid nanomaterials not only decreased neuronal death/damage but also increased biocompatibility [101,102]. 

FDA-approved superparamagnetic iron oxide nanoparticles were used to transport mesenchymal stem cells to retinal tissue. The mesenchymal stem cells were first magnetized with superparamagnetic iron oxide nanoparticles and were then injected intravitreally into a rat model of retinal degeneration in the presence or absence of a gold-plated neodymium magnet placed outside the eye. The placement of the magnet outside the eye resulted in a 10-fold increase in the number of deposited mesenchymal stem cells, with little or no damage to the eye compared to unlabeled cells. Magnetic mesenchymal stem cell treatment with an orbital magnet also led to a subsequent increase in the expression of interleukin-10 (an anti-inflammatory cytokine) [103]. Researchers have also used magnetite nanoparticles and magnetic force to coculture a heterogeneous layer of cells and termed it as “magnetic force-based tissue engineering” [53]. The use of magnetic force-based tissue engineering technology enhanced the proliferation of human retinal pigment epithelial cells within 24 h of culture [104]. Fluorescent polydopamine (PDA) nanoparticles are widely used nanoparticles in medical imaging and sensing due to their adjustable fluorescent properties, biocompatibility, and biodegradability [105]. Recently, a study demonstrated the free radical scavenging effect of polyhedral oligomeric silsesquioxane (POSS)-based polyphenol nanoparticles in the NIH-3T3 cell line and suggested its potential use in oxidative-stress-induced pathologies [106,107].

### 2.4. Nanoscaffolds

Nanoscaffolds are self-assembled or electrospun nanofibers made up of synthetic or natural polymers. Nanoscaffolds provide a microenvironment for cellular signaling that influences the proliferation, migration, and differentiation of various cells [108]. Electrospun nanofibers are made up of microscopic tubes (100–200 nanometers in diameter). Electrospinning is helpful for intertwining these microscopic tubes to form a web. In the field of tissue engineering and regenerative medicine, designing biocompatible cellular scaffolds is a recent trend. The nanoscaffolds are designed to simulate the structural features and pattern of the extracellular matrix. There are three types of nanoscaffolds: natural nanoscaffolds, synthetic nanoscaffolds, and hybrid nanoscaffolds.

These scaffolds are made up of natural nanofibers/polymers. Collagen I is a major component of retinal pigment epithelial cells, and therefore, ultrathin collagen I membranes were used to design natural nanoscaffolds. These membranes were stable for 10 weeks and degraded within 24 weeks. Other natural polymers used for retinal regeneration studies include gelatin [73], fibrin [74], chitosan [75], laminin [76], and hyaluronic acid [77]. The chemistry of natural nanoscaffolds makes them more suitable for cell attachment and biological activity [78].

Synthetic nanoscaffolds are easier to design, and their physical properties can more easily be controlled to mimic the extracellular matrix compared to natural polymers [90]. Poly (lactic-*co*-glycolic acid) (PLGA) [79], poly (ε-caprolactone) (PCL) [80], poly (L-lactic acid) (PLA) [76,81], polyimide [82], and poly (l-lactide-co-ε-caprolactone) [79] are commonly used synthetic polymers. 

Biohybrid nanoscaffolds are made by combining both natural and synthetic nanofibers to make composite scaffolds. Biohybrid nanoscaffolds have the appearance and protein composition of a natural nanoscaffold and the design of synthetic nanoscaffolds. Studies have shown that biohybrid nanoscaffolds are well tolerated without any adverse inflammatory reaction in the retina [83], but there is a need to characterize the various components of biohybrid nanoscaffolds for their reproducibility.

## 3. Studies on the Application of Nano-Biomaterials for Retinal Regeneration

Retinal transplantation is considered a limiting factor for the treatment of blinding diseases due to the complex neural network [109]. Therefore, tissue regeneration using scaffolds with acceptable biocompatibility is a recent, more promising approach to repair damaged tissues or organs. Scaffolds likely simulate the extracellular matrix (ECM) and thus have the capability to support cell migration, adhesion, and morphology in the regeneration of the retina [109,110,111]. Nanomaterials with unique properties and a hierarchical architecture have been developed for multidisciplinary applications and have the capability to significantly advance the field of tissue/organ regeneration. As a result, various investigators have developed nanomaterials with better biocompatibility, electroconductivity, and cell adhesion to enhance the efficiency of tissue regeneration. [112,113,114,115,116]. Various in vitro, in vivo, and therapeutic studies have highlighted the importance of nanostructures in retinal regeneration, and a summary is presented in Table 2.

### 3.1. In Vitro Studies on Nano-Biomaterial Implantation and Imaging

The use of established in vitro approaches has helped researchers to make rapid and reliable assessments of nanomaterials’ health hazard potential. Furthermore, in vitro toxicological studies of nano-biomaterials help us to understand the interactions of these engineered nanomaterials at the cellular level. For instance, the proliferation, migration, and differentiation of retinal progenitor cells (RPCs) have been evaluated by seeding them on smooth poly (ε-caprolactone) and short (2.5 μm) and long (27.5 μm) poly (ε-caprolactone) nanowire scaffolds. Scanning electron microscopy (SEM) analysis revealed that individual RPCs create cell-to-cell contacts with lamellipodia-like structures. However, they maintained their spheroid shape when seeded on small and large poly (ε-caprolactone) nanowires, whereas they showed no distinctive morphologic changes on smooth poly (ε-caprolactone) nanowire scaffolds. Immunohistochemistry studies showed an increased expression of protein kinase C (bipolar cell marker) and recoverin (photoreceptor marker) when RPCs were cultured on small and large poly (ε-caprolactone), except for smooth poly (ε-caprolactone) scaffolds, indicating the presence of a differentiated cell population [54]. Nanowire arrays of gallium phosphide (0.5–4 mm) with different topographies enabled the development of long-term in vitro cultures of postnatal retinal cells including ganglion cells, photoreceptors, and bipolar cells. Scanning electron microscopy analysis revealed that gallium phosphide nanowires facilitate the growth of cells of different sizes and morphologies. Immunocytochemistry and fluorescence microscopy analyses showed that unlike flat gallium phosphide substrate, gallium phosphide nanowires provided a better attachment for neurons to extend and form branched neurites. These results indicate that gallium phosphide nanowires exhibit excellent substrate properties for retinal cells compared to short nanowires and flat controls [61]. Conversely, short- and long-term culturing of mouse retinal cells on silicon nanowires (length: 4.4 μm; diameter: 20 to 120 nm) showed a strong adherence of densely packed single-cell layers and the absence of cell clusters, similar to cell cultures grown on flat silicon substrates [94]. A decreased expression of retinal markers was observed in cells grown on both types of substrates [95]. Furthermore, the functionalization of silicon nanowires with perfluorosilane molecules not only prevented the striking changes in cell phenotype but also limited the exposure to contaminants [56].

Researchers are focusing on the use of gold nanoparticles to improve ophthalmic diagnosis and therapy. Due to their antiangiogenic and anti-inflammatory properties, low or no cytotoxicity, inert nature, plasmon band, and biocompatibility, gold nanoparticles are useful not only for diagnosis but also for ocular therapeutics [76,116]. Recently, the in vitro biocompatibility of gold nanoparticles with retinal pigment epithelial (ARPE-19) cells was evaluated using 2D and 3D confocal imaging. The results indicate that gold nanoparticle internalization and biocompatibility depend on the shape and the size of the nanoparticles [45]. Likewise, Hayashi et al. also did not observe any inhibitory effect of gold nanoparticles on ARPE-19 cell proliferation [91]. To develop a cell-based therapy for retinal degeneration, human Wharton’s Jelly derived mesenchymal stem cells were loaded with gold nanoparticles (80 nm), which resulted in little cell death over a period of 10 days compared to the control. The histological studies provided evidence that the transplantation of human Wharton’s Jelly derived mesenchymal stem cells in subretinal space delayed retinal degeneration with no systemic migration and retinal tumorigenesis. Furthermore, confocal microscopy showed the presence of human Wharton’s Jelly derived mesenchymal stem cell markers in Müller cells, photoreceptors, and bipolar cells [118].

A detailed assessment of silver and gold nanoparticle uptake and toxicity was performed using an in vitro tissue culture model depicting the mouse retina. Analysis of the retina using transmission electron microscopy revealed the presence of silver and gold nanoparticles in all neuronal layers of the retina. This model is advantageous for testing the neurotoxicity of nanoparticles in a controlled and defined serum-free culture medium. In addition, this particular study demonstrated that even minimum concentrations of nanoparticles have adverse effects on neural tissue [46]. The antiangiogenic effect of gold nanodisks (20 nm thick and 160 nm in diameter) was evaluated using in vitro angiogenic assays. The gold nanodisks were shown to attenuate VEGF-induced human retinal microvascular endothelial cells’ migration at 3 pM concentrations, but this inhibition was less effective at 1 pM. However, the endothelial cells did not exhibit any toxic effects from gold nanodisks even at higher concentrations (~10^4^ per cell) after 48 h of incubation [56].

Natural nanoscaffolds made up of ultrathin (7 µm) collagen I membranes were a viable substrate for subretinal implantation and retinal pigment epithelium (RPE) cell attachment [119]. In some studies, alginate beads and cross-linked gelatin/chitosan scaffolds were used for RPE cell maintenance and proliferation [75,120]. 1-Ethyl-3-(3-dimethyl aminopropyl) carbodiimide cross-linked gelatin scaffolds were found to be stable and suitable for retinal sheet implantation [121]. Decellularized retinal scaffolds are also used for human retinal progenitor cell attachment and growth [122]. PCL scaffolds are the thinnest biodegradable scaffolds available with no adverse pathological effects [123]. The adherent capacity of human RPE cells was improved when the surface morphology of the PCL scaffolds was modified to decrease the hydrophobicity of PCL by alkaline hydrolysis [124]. The coating of PCL scaffolds with xeno-free synthetic arginine–glycine–aspartic acid (RGD) peptides induced the differentiation of rods, but the construct was found to be inappropriate for human use due to the high expression of stem cell markers [125]. Collagen IV-coated porous honeycomb PLA films were able to increase the proliferation and survival of human embryonic stem cells. Various studies have shown that PCL–gelatin scaffolds, PDMS–laminin scaffolds, and PLGA–collagen type I scaffolds promote the growth and functional maturation of RPE cells [79,126,127,128,129]. In addition to natural and synthetic nanoscaffolds, biohybrid nanoscaffolds were also evaluated for retinal cell growth and differentiation. Various combinations of PLLA and PLGA were tested by Thomson et al. for RPE cell growth, and the combination with a 25:75 PLLA/PLGA ratio was found to have the lowest cell death [130]. A biohybrid scaffold made of poly (L-lactic acid-co-ε-caprolactone) and silk fibroin at a 1:1 ratio promoted proliferation and differentiation of RPCs into photoreceptors [131]. 

For the repair and reformation of damaged retinal tissues, phosphor nanoparticles have been synthesized by sol–gel method, coated with PEG, further dispersed in CS-PCL copolymer, and electrospun to form SrAl2O4: Eu2+, Dy3+/CS-PCL scaffolds. SrAl2O4: Eu2+, Dy3+/CS-PCL (30%) electrospun scaffolds showed better cytocompatibility, excellent proliferation rates, and adequate differentiation towards mice retinal progenitor cells, demonstrating their suitability for curing retinal diseases [132]. Core–shell fibrous scaffold from polyethylene glycol/polycaprolactone, encapsulated with retinal pigmented epithelium-derived factor (PEDF), has been employed for differentiation of conjunctiva mesenchymal stem cells (CJMSCs) into photoreceptor-like cells. CJMSCs showed higher expression of rhodopsin when seeded on PEDF-loaded PEG/PCL scaffold, compared to cells cultivated on tissue-culture polystyrene [133]. PEGylated graphene oxide (PEG-GO) with high oxidation levels exhibited a higher degree of (ROS)-dependent cytotoxicity in human retinal capillary endothelial cells and human corneal epithelial cells [134].

### 3.2. In Vivo Studies on Nano-Biomaterial Implantation and Imaging

The regenerative capacity of the mammalian retina is very low; therefore, subretinal implantation of retinal progenitor cells (RPCs) provides a higher restorative potential for retinal diseases. In various animal models, RPCs have shown the ability to migrate, differentiate into mature photoreceptors, and form synapses with existing retinal cells [135]. Various modifications of poly (ε-caprolactone) (PCL) scaffolds, such as smooth PCL scaffolds, electrospun PCL, and PCL short nanowires, were subretinally implanted in pigs. The implantation of these scaffolds resulted in limited tissue disruption and no inflammation. A 12 mm^2^ membrane was inserted in the visual streak area and the electrophysiological effects of all three PCL scaffolds were tested using infrared (IR) fundus imaging and multifocal electroretinography (mfERG). Out of all the scaffolds tested, the PCL short nanowire was observed to be the most suitable candidate for subretinal implantation [61].

Furthermore, subretinal implantation of an artificial photoreceptor, composed of an oriented gold-nanoparticle-decorated titania (Au-TiO_2_) nanowire array, evoked light activities in the primary visual cortex of blind mice. After surgical subretinal implantation of nanowire arrays, their interface with the retina was demonstrated using scanning electron microscopic images, exhibiting their potential application in retinal prosthesis. The population light responses estimated via calcium-sensitive protein (GCaMP6s) imaging indicated that the nanowire-array-interfaced retina restored light responses in blind mice. Additionally, the retina and retinal ganglion cells of blind mice were found to be unaffected by the implant even 5 months after the nanowire array implant surgery. The infrared images obtained 4–8 weeks after nanowire array implantation showed a comparable level of population light responses between wild-type and blind mice, suggesting the recovery of light sensitivity in many colors [14]. Intravitreal injection of gold nanodisks (20 nm thick and 160 nm in diameter) has also been shown to attenuate retinal neovascularization in a murine model of oxygen-induced retinopathy. Immunostaining of the retina revealed that gold nanodisks inhibit neovascularization in a dose-dependent manner. Furthermore, histological observations do not show any signs of inflammation or disruption in retinal integrity. Apoptosis determined using terminal deoxynucleotidyl transferase dUTP nick end labeling (TUNEL) assay showed no signs of apoptosis after the administration of gold nanodisks. Optical coherence tomography (OCT) images after intravitreal injection showed a diffused distribution of gold nanodisks in the vitreous at as early as 6 h and subsequent clearance by 2 weeks, indicating that there was no drastic change in retinal function or integrity [47].

The blood–retinal barrier is a tightly regulated physiologic barrier that isolates neurons from the blood. Interestingly, intravenous administration of gold nanoparticles (1 g of Au NP/kg of mice) showed that they pass through the blood–retinal barrier without causing any cytotoxicity in mice. The use of transmission electron microscopy showed the distribution of gold nanoparticles in all the retinal layers after 24 h of 20 nm gold nanoparticle injection, indicating their passage through the blood–retinal barrier. Histological and TUNEL analyses showed no abnormal effect of nanoparticle movement on relative retinal thickness or retinal cell apoptosis. In addition, immunohistochemistry analysis showed no change in the expression levels of zonula occludens-1 or glut-1 in the retina [44]. Intravitreal injection of gold nanoparticles (670 μmol/0.1 mL) in Dutch belted rabbits showed no signs of toxicity in the optic nerve and retina even 1 month after injection [43]. In recent years, gold nanorods coated with anti-CD90.2 antibodies or poly (strenesulfate) have been intravitreally injected in mice to evaluate their efficiency as a contrast agent for ocular OCT. Images obtained using high-magnification transmission electron microscopy showed the presence of gold nanorods in the vitreous. Furthermore, the presence of gold nanorods obscured the retinal signal and induced ocular inflammation, limiting the use of gold nanorods as contrast agents with OCT [57]. Recently, gold nanoparticles were introduced as image enhancers for multimodal tracking of transplanted photoreceptor precursors by OCT and computed tomography. Fluorescently labeled photoreceptor precursors and gold nanoparticles were transplanted in the subretinal and vitreal space of 4–8-week-old Long–Evans pigmented rats and longitudinally monitored for up to 1 month using computed tomography, OCT, and fluorescence fundus imaging. Fluorescence imaging did not indicate intraocular inflammation or other toxic effects on the retina and vitreous space. The OCT images showed subretinal colocalization of the nanoparticle-labeled photoreceptor precursors. In addition, the use of gold nanoparticle labeling enabled visualization of the photoreceptor precursors in subretinal space using computed tomography imaging [117].

The visual system of mammals detects light between 400 and 700 nm but is unable to see near-infrared (NIR) light. Recently, a self-powered, injectable photoreceptor-binding NIR nanoantenna has been reported, which can enhance the visual spectrum of mammals to the NIR range. Nanoantenna/upconversion nanoparticles (UCNPs) are composed of core–shell-structured upconversion nanoparticles. To develop water-soluble photoreceptor-binding upconversion nanoparticles (pbUCNPs), concanavalin A protein was conjugated with polyacrylic acid coated upconversion nanoparticles. These upconversion nanoparticles/nanoantennae bind to the photoreceptors when injected into the subretinal space. Histological and TUNEL analyses showed little or no side effects 3 days/10 weeks after upconversion nanoparticle injection in the retina. Optical images of mouse pupils injected with photoreceptor-binding upconversion nanoparticles showed strong pupillary constrictions when exposed to near-infrared light. Interestingly, these implanted upconversion nanoparticles/nanoantennae were biocompatible and showed no obstruction with normal visible light vision [15]. Figure 4 shows the surface modifications of pbUCNPs and their distribution in mice retina. 

The implantation of nanostructured PCL thin films in rabbit eyes exhibited good tolerance and structural integrity of design features over 9 months of ocular residency. No ocular inflammation or adverse cellular reactions were observed in rabbit eyes [136]. Another study showed that ultrathin poly (methyl methacrylate) (PMMA) scaffolds with pores not only provide a suitable cytoarchitectural environment to retain retinal progenitor cells, but it is also easy to implant the scaffold to a specific retinal region [137]. A study by Thomas et al. has shown that RPE cells on the parylene-C scaffold remained as a monolayer up to 21 weeks post-implantation in the subretinal space of rats [138]. Furthermore, subretinal implantation of CPCB-RPE1 (California Project to Cure Blindness-Retinal Pigment Epithelium 1) on parylene-C scaffolds in Yucatán minipigs showed that the RPE cells survived and remained intact as a monolayer for nearly one month [139].

Methylprednisolone nanoscale zirconium–porphyrin metal–organic framework nanoparticles have been synthesized using nanoscale zirconium–porphyrin metal–organic framework (NPMOF) loaded with methylprednisolone (MPS), a common drug for retinal degenerative diseases. Intraocular administration of MPS-NPMOF nanoparticles in the injured retina of adult zebrafish resulted in faster photoreceptor regeneration with excellent in vivo biocompatibility and low biotoxicity [140]. The drug bioavailability and its retention in the eye is a major issue for the treatment of ocular diseases related to the posterior segment of the eye. The silk fibroin nanoparticles (SFNs) encapsulated with fluorescein isothiocyanate labeled bovine serum albumin (FITC-BSA) showed excellent biocompatibility, better retention, and no cytotoxicity in human retinal pigment epithelial cell lines. Furthermore, intravitreal administration of FITC-BSA-SFNs in New Zealand white rabbits demonstrated prolonged retention, enhanced drug bioavailability/delivery, and constant therapeutic efficiency compared to the FITC-BSA solution [141]. An approach for selective drug delivery to injured sites was demonstrated using maghemite magnetic nanoparticles (MNPs) covalently conjugated with nerve growth factor (NGF) complex. The in vitro differentiation of PC12 cells on the 3D platform after adding NGF-MNPs clearly demonstrated the uptake of NGF growth factors by the cells. Likewise, in vivo administration of NGF-MNPs to the mouse sciatic nerve and retina showed the accumulation of these carriers at the external magnet site, thereby demonstrating the penetration efficacy, biocompatibility, and stability of the magnetic nanocarriers [142].

### 3.3. Therapeutic Studies on Nano-Biomaterial Implantation and Imaging

Currently, intravitreal and subretinal injections are the only two approaches available for the administration of therapeutic agents to treat retinal diseases. However, these methods have several drawbacks or are ineffective; therefore, the development of effective therapeutics is consequently a primary ophthalmic research goal [48,117,143]. A plethora of studies in the biomedical field have harnessed nanotechnology to reformulate and revolutionize the use of therapeutic agents. However, the use of nanostructures as therapeutic agents for the treatment of ocular pathologies is still in its infancy stage.

In recent years, researchers have mainly focused on developing therapeutic agents that can directly target signaling molecules involved in angiogenesis. For instance, Song et al. reported the use of 160 nm sized gold nanodisks that not only bind with vascular endothelial growth factor (VEGF) but also inhibit VEGF-induced angiogenesis in human retinal microvascular endothelial cells. They have also shown that intravitreal injection of gold nanodisks inhibits aberrant retinal angiogenesis in a mouse model of oxygen-induced retinopathy [47]. Furthermore, gold nanoparticles (3–5 nm) inhibit VEGF-induced migration of choroid retina endothelial cells (RF/6A) by inhibiting Akt/eNOS signaling pathways [42]. The use of silver nanoparticles (40–50 nm) also inhibits VEGF-induced migration, proliferation, and tube formation of bovine retinal endothelial cells by inactivating PI3K/Akt signaling pathways [58]. Similarly, the use of silver nanoparticles inhibits VEGF-induced matrigel plug angiogenesis in mice [58]. Silver nanoparticles attenuate VEGF- and IL-1beta-induced porcine retinal endothelial cell permeability via inactivation of the Src kinase pathway. Silver, being a cost-effective material, might be considered a good candidate for the treatment of various proliferative retinopathies [49].

Superparamagnetic iron oxide nanoparticles can be used to develop a new strategy for cell therapeutics. These nanoparticles are in use to guide intravenously injected rat mesenchymal stem cells to a specific retinal locus. Furthermore, magnetic mesenchymal stem cell treatment leads to an increase in anti-inflammatory molecules, suggesting a potential implication of mesenchymal stem cell-based treatment in clinical therapies [103]. A recent study by Maya-Vetencourt et al. demonstrated that subretinal injection of conjugated polymer (poly (3-hexylthiophene), P3HT) nanoparticles mediated light-stimulus-induced stimulation in photoreceptors and rescued vision in a rat model of retinitis pigmentosa [50]. They also observed a stable and wide coverage of these P3HT nanoparticles in the retina, as shown in Figure 5. 

Several studies have utilized stem cell-derived RPE-based therapies in patients with age-related macular degeneration (AMD). In a recent study, a PLGA nanoscaffold was used to deliver clinical-grade AMD-patient-derived induced pluripotent stem cell (iPSC) RPE in rodent and porcine laser-induced RPE injury AMD models [51]. As shown in Figure 6, the iPSC-RPE patches were safe, and once the biodegradable PLGA scaffold degraded, the AMD-iRPE patch was integrated on Bruch’s membrane and was fully functional. In another study, an amniotic membrane was used to support RPE cells and substitute Bruch’s membrane in a pig model of injury-induced choroidal neovascularization [144]. In addition, various studies have used RPE and parylene-C implants to rescue vision loss in preclinical and translational studies [138,145,146].

## 4. Challenges and Future Perspectives

Nano-biomaterials’ safety, stability, and biocompatibility are the challenging issues for their application in retinal regeneration and repair. Therefore, more advancements in the field of nanotechnology are required for the development of biocompatible nanomaterials with superior physicochemical properties and better retention ability. Recently, with the advancement in semiconductor technology, the implantation of an electronic chip device that mimics cellular functions and treats retinopathies has become possible. Given this, a study showed that micropatterned graphene oxide (GO), when integrated with a retinal prosthesis, enabled the adhesion of retinal cells. In addition, GO-based nanomaterials provide enhanced safety, improved tissue regeneration, and repair [147]. Ranibizumab is an active humanized monoclonal antibody that neutralizes vascular endothelial growth factor A. Yan et al. synthesized a formulation based on ranibizumab-conjugated iron oxide (Fe_3_O_4_)/PEGylated polylactide-co-glycolide (PEG-PLGA) for the treatment of neovascular age-related macular degeneration. The in vitro analysis revealed that Fe_3_O_4_-loaded PEG-PLGA polymer nanomaterial showed considerable antiangiogenic activity. Moreover, ranibizumab-conjugated PEG-PLGA exhibited a negligible effect on the proliferation of human endothelial cells [148]. Chittasupho et al. developed a dual-action drug delivery system based on R5K peptide and itraconazole (ITZ) that blocked VEGF binding to its receptor and altered its signaling pathway. The itraconazole-encapsulated PLGA nanoparticles, conjugated with R5K peptide, enhanced the antiangiogenic effects of drugs used for AMD patients. In in vitro studies, R5K-ITZ-NPs demonstrated high potential, showing cell-specific and dose-dependent inhibition of vascular endothelial cell migration, proliferation, and tube formation in response to VEGF stimulation [149].

Recently, a biocompatible, biodegradable, and injectable photoresponsive nanosystem (HA-NSP) was developed for AMD and diabetic retinopathy. HA-NSP was synthesized using azoprolamin (AZP) nanospheres and then incorporated in a hyaluronic acid (HA) hydrogel. Dynamic light scattering (DLS) analysis revealed that AZP nanospheres with a particle diameter of 94 nm showed photoresponsiveness to UV = 365 nm. HA-NSP demonstrated encapsulation efficiency of 85%, and up to 60% of IgG was released over 32 days. Furthermore, HA-NSP showed no cytotoxicity in retinal pigment epithelium cell lines [150]. Jung et al. developed a neural stimulation device based on silicon nanowire (siNW) as a photodetector and used field-effect transistors for neuronal activation. The neural stimulation device has a unit cell size of 110 × 110 μm and a resolution of 32 × 32 on a flexible film with a thickness of approximately 50 μm. In vitro studies involving the retinal tissue of retinal degeneration 1 (rd1) mice demonstrated the usefulness of the proposed device as a retinal stimulation device. Furthermore, the application of stimulation pulses to the retinal tissues effectively showed a significant increase in neural response signals in proportion to light intensity [151].

Despite emerging avenues for nanostructures in biomedicine, there is a need to carry out systematic research to overcome various challenges. Retinal impairment is a leading cause of irreversible vision loss due to various ophthalmic diseases. Various in vitro investigations have shown no toxic effects of nanoparticles and nanowires; however, long-term in vivo studies are critical for understanding the importance of nano-biomaterials in regenerative medicine. The current use of nano-biomaterials for eye treatments has prevented and cured various retinopathies. However, these nanostructures are delivered using invasive procedures, resulting in the possibility of triggering inflammation, infection, or retinal detachment, which leads to vision loss. These limitations not only reduce the patient’s quality of life but also impact our health care system economically. Therefore, more pharmacokinetic studies, pharmacodynamic studies, and toxicological studies are needed to understand the bioeffects of nano-biomaterials on the retina. Advancements in the nano-biomaterial engineering field will help us in discovering new approaches for retinal diagnosis and therapy. Noble metals, bioinspired magnetic nanoparticles, and nanowires represent powerful tools for real-time imaging to overcome the drawbacks of standard treatments.

Hybrid nanostructures are useful as they carry only merits while overpowering the demerits of nanomaterials. Furthermore, hybrid nanostructures penetrate deeper inside the ocular segments, resulting in the localized delivery of drugs at high dosages, enriching the local drug concentration. Recently, a few studies have shown that nanoparticle-based eye drop formulations can deliver apatinib and triamcinolone acetonide to the rat retina [152,153]. Therefore, improvement of the noninvasive infiltration of nanomaterials through the blood–retinal barrier will be a stepping stone in the field of biomedicine. An alternative approach is to incorporate stem cells with hybrid nanostructures to initiate the formation and further regeneration of the retina. The implication of hybrid nanostructures will be a state-of-the-art future systemic strategy to overcome the challenges between in vitro and in vivo studies. Hybrid nanostructures can regulate neural signals and further simulate those signals into the extracellular matrix. Such a combination could assist in conveying the signals to the brain to form clear vision. Hence, the integration of hybrid nanostructures with stem cells could also be a bottom-up approach for retinal rejuvenation in forthcoming investigations.

## 5. Method of Literature Search

For this review, the researchers conducted a MEDLINE/PubMed search for articles published between 2000 and 2020, using the following keywords: “nanoparticles”, “nanodisks”, “scaffolds”, “nano biomaterials & retina”, “nano-scaffolds & retinal regeneration”, and “nanoparticles & retinal regeneration”. Current contents and relevant articles on the role of nanoparticles or nanomaterials in retinal regeneration were also obtained using a Google search. Published papers in languages other than English were excluded. We read all 153 articles and did not contact any authors.

## Figures and Tables

**Figure 1 nanomaterials-11-01880-f001:**
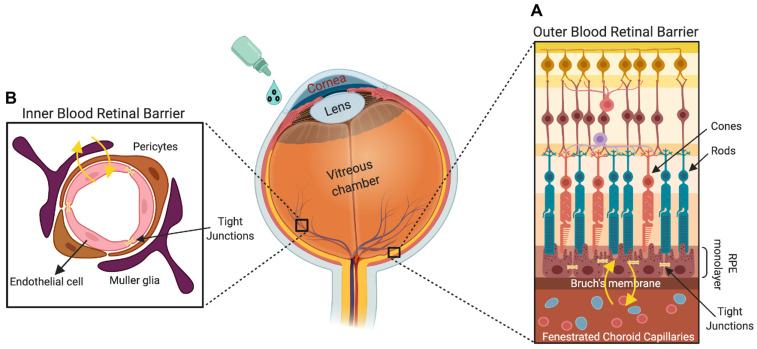
The retina is the innermost multilayered light-sensitive region of the eye, which has multiple barrier systems to regulate molecule, ion, and water flux out of the retina. The blood–retinal barrier is composed of two components: (**A**) the outer blood–retinal barrier, which comprises tight junctions that regulate paracellular trafficking between pigment epithelial cells, and (**B**) the inner blood–retinal barrier (comprising Müller cells, pericytes, and endothelial cells) formed by tight junctions that plug neighboring capillary endothelial cells. The tight junctions between retinal pigment epithelial and capillary endothelial cells form the outer and inner blood–retinal barriers, respectively, and these barriers regulate the retinal homeostasis and visual function.

**Figure 2 nanomaterials-11-01880-f002:**
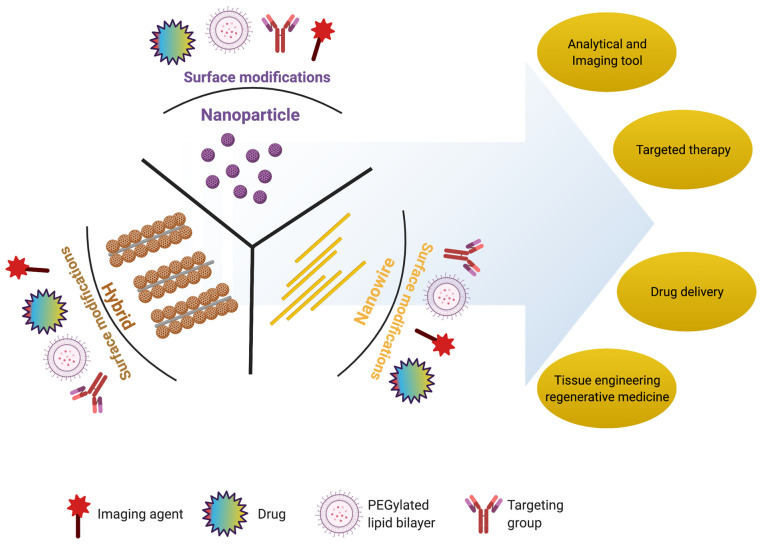
Schematic representation of multifunctional nanostructures: nanoparticle (NP), nanowire (NW), and hybrid with various applications in biomedical science. These nanostructures can be surface modified with drugs (incorporated or conjugated to the surface), a PEGylated lipid bilayer (to improve solubility and decrease immunogenicity), targeting groups (to improve nanostructures’ circulation, effectiveness, and selectivity), and imaging agents (e.g., fluorescent dyes used as reporter molecules and employed as tracking or contrast agents).

**Figure 3 nanomaterials-11-01880-f003:**
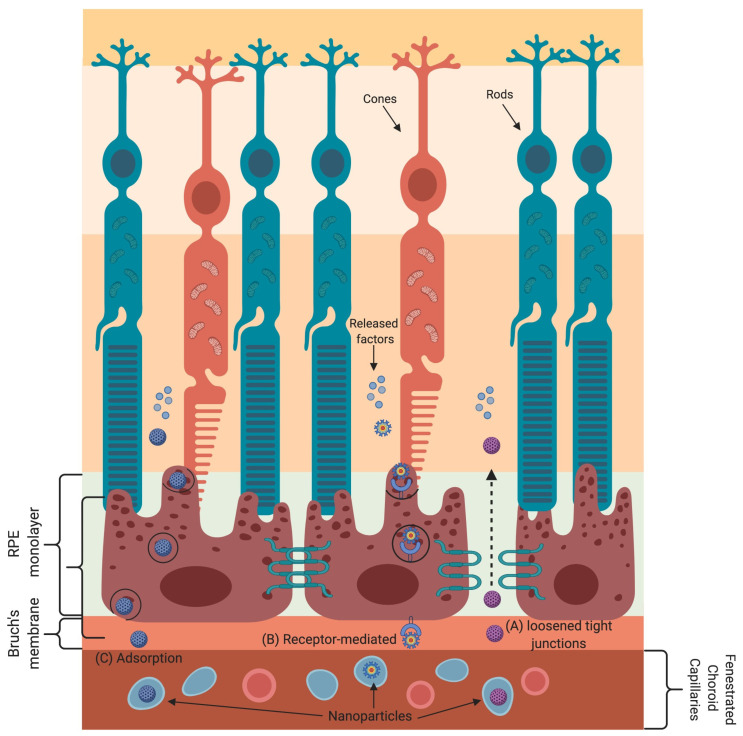
Transport mechanisms for delivering nanoparticles (NPs) into the blood–retinal barrier (BRB). The BRB is exceedingly selective and has unambiguous transport mechanisms allowing a close control of molecules/cells that enter the retina. Loosening of tight junctions (TJs) either due to the presence of a surfactant in NPs or by BRB impairment due to pathological conditions allows the movement of NPs through the BRB. (**A**) NPs’ admittance into the retina is through receptor-mediated transcytosis. (**B**) The NPs interact with respective receptors on the endothelial cell surface, which leads to plasma membrane invaginations, vesicle formation, and, therefore, the release of the NPs at the other side of the membrane. (**C**) NPs coated with chitosan or other polysaccharides can cross the BRB by adsorptive transcytosis.

**Figure 4 nanomaterials-11-01880-f004:**
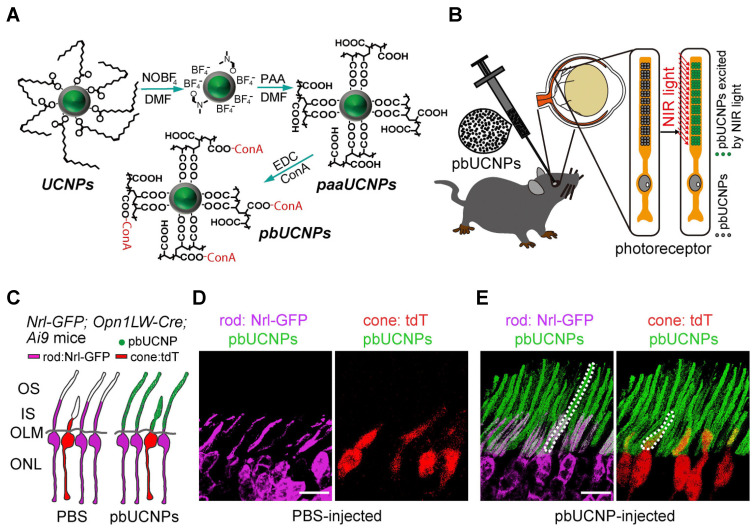
Photoreceptor-binding upconversion nanoparticles (pbUCNPs) and their distribution in the retina. (**A**) The surface modification for photoreceptor-binding UCNPs (pbUCNPs) is presented. (**B**) Schematic diagram to show the subretinal injection and binding of pbUCNPs to the photoreceptor outer segments and green light generation after exposure to near-infrared (NIR) light. (**C**) Schematic diagram to show pbUCNP (green) distribution in the retina. The cones were labeled with Opn1LW-Cre;Ai9-lsl-tdTomato (pseudo color red), and the rods were labeled with Nrl-GFP (pseudo color violet). (**D**,**E**) The mice were injected with PBS (**D**) and pbUCNP, and the self-anchored pbUCNPs (green) are presented as dashed lines in both the inner and outer segments of rods (violet) and cones (red). The outer nuclear layer is abbreviated as ONL; the outer limiting membrane is abbreviated as OLM; the inner segment of the photoreceptors is abbreviated as IS; the outer segment of the photoreceptors is abbreviated as OS. Reprinted with permission from [15]. Copyright Elsevier, 2019.

**Figure 5 nanomaterials-11-01880-f005:**
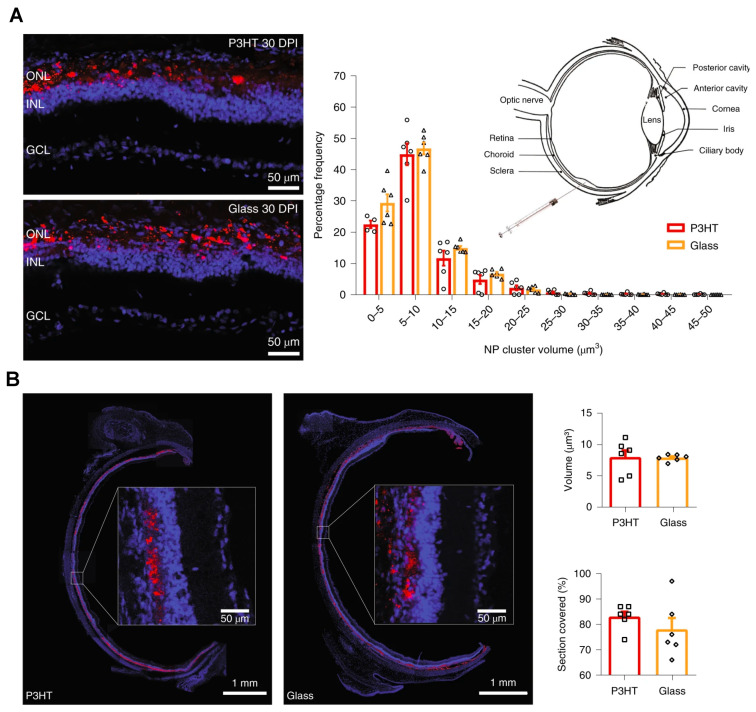
Wide and stable coverage of subretinally injected P3HT nanoparticles in the retina. (**A**) The Royal College of Surgeons (RCS) rats were injected with fluorescent glass nanoparticles or P3HT NPs at 30 DPI. The image shows the presence of NP fluorescence (red) with bisbenzimide nuclear staining (blue). *Z*-stack confocal retinal section images were used to estimate the percentage frequency distribution of the volume of NP clusters at 30 DPI. *p* > 0.10, two-sided Kolmogorov–Smirnov test. (**B**) A full equatorial reconstruction of RCS rat retinas injected with the indicated NPs at 30 DPI showed the NP (red) diffusion in the whole subretinal space, with no distribution or penetration in the internal retinal layers (insets at higher magnification). The bar graph shows the total NP volume in the subretinal space and the percentage coverage of retina by NPs at 30 DPI. *p* > 0.05, two-sided Mann–Whitney *U*-test. The bar graphs represent mean ± s.e.m. of *n* = 6 RCS rats for each experimental group. Reprinted with permission from [50]. Copyright 2020 Springer Nature.

**Figure 6 nanomaterials-11-01880-f006:**
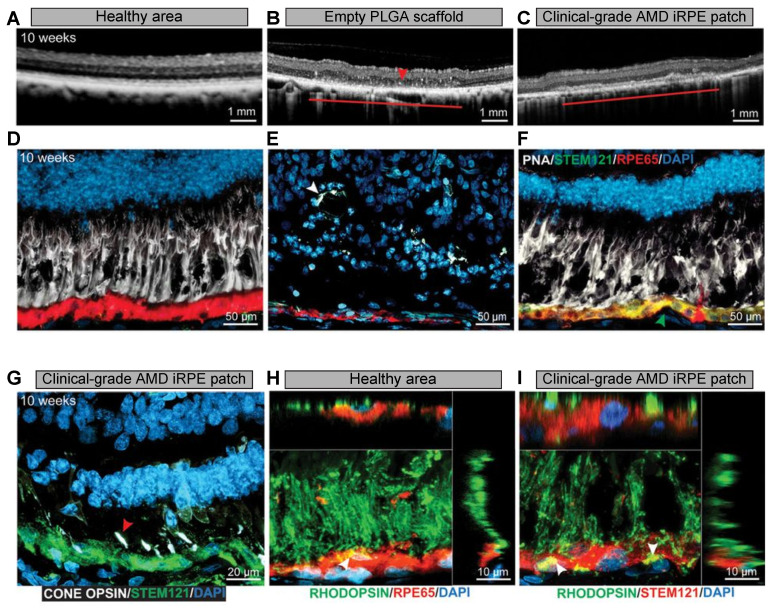
Integration and functional assessment of AMD-patient-derived iRPE patch in laser-induced porcine retinal degeneration model. (**A**–**C**) OCT images of healthy retina and retina transplanted with PLGA scaffold or AMD-iRPE patch. The horizontal line shown in (**B**,**C**) indicates the transplanted scaffold over an area of laser-induced RPE ablation. (**D**–**F**) The retinal sections were stained with PNA (white), STEM121 (green), and RPE65 (red). The green arrow in (**F**) indicates the integration of AMD-iRPE (green, STEM121) in the injured porcine retina (red, RPE65). The white arrowhead in (**E**) shows the presence of retinal tubulations. (**G**) The retinas were immunostained for red, blue, and green cone opsins (white) and STEM121 (green) to observe the presence of pig cone receptors above human iRPE (red arrow). (**H**,**I**) The retinas were immunostained with rhodopsin and RPE65 or STEM121. The white arrow shows the presence of phagocytosed photoreceptor outer segments in both the healthy pig RPE (stained with RPE65) and in human iRPE cells (STEM121). Z-sections show the localization of photoreceptor outer segments. Reprinted with permission from [51]. Copyright The American Association for the Advancement of Science, 2019.

**Table 2 nanomaterials-11-01880-t002:** In vitro and in vivo studies of various nanomaterials and nanoscaffolds used for retinal regeneration.

Analysis	Nanomaterial	Form	Size (nm)	Cell Response	Ref.
In vitro	Poly (ε-caprolactone) (PCL)	NWs	Length: 2500	↑ expression of PKC and recoverin in RPCs; cells undergo differentiation	[54]
	Gallium phosphide (GaP)	NWs	Length: 500–4000	Extended growth of retinal cells	[61]
	*n*-type silicon	NWs	Length: 440	Long-term and dense growth of mouse retinal cells	[95]
	Gold (Au)	Nanoparticle	Diameter: 5–100	ARPE-19 cells undergo apoptosis upon AuNP internalization	[77]
			Diameter: 10–12	Gold nanoparticles inhibit proliferation of ARPE-19 cells; no cytotoxicity	[16]
			Diameter: 80	Highly viable mesenchymal stem cells undergo differentiation and secrete various trophic factors	[15]
	Gold (Au), silver (Ag)	Nanoparticle	Diameter: 20–80	Increase uptake into retinal cells; ↑ apoptosis, oxidative stress, and microglia activation	[58]
	Gold (Au)	Nanodisk	Diameter: 160	Inhibition of in vitro angiogenesis without cellular toxicity of HRMECs	[56]
	Hybrid nanoscaffolds	Combination of *Antheraea* *pernyi* silk fibroin (RWSF), PCL, and gelatin	Diameter/porosity: 90–210	Increased expression of RPE marker genes (CRALBP, PEDF, VEGF, MITF, and PMEL 17 among others)	[83]
In vivo	Poly (ε-caprolactone) (PCL) membranes	NWs	Length: 2500	Successful implantation into subretinal space with limited tissue disruption and no inflammation	[54]
	Gold (Au), titania (TiO_2_)	Au nanoparticle coated TiO_2_ NWs	AuNPs diameter: 5–15, TiO_2_ NW length: 2000	AuNP-decorated TiO_2_ NW arrays restore light-sensitive visual responses in degenerated photoreceptors	[14]
	Gold (Au)	Nanodisk	Diameter: 160	Intravitreal injection attenuates neovascularization in mouse model of oxygen-induced retinopathy	[56]
	Gold (Au)	Nanoparticle	Diameter: 20–100	Intravitreal injection of gold nanoparticles passed through the blood–retinal barrier with no structural abnormality or cell death	[91]
	Gold (Au)	Nano-gold	Not reported	No retinal or optic nerve toxicity by intravitreal injection of nano-gold	[43,91]
	Gold (Au), poly (strenesulfate)	Poly (strenesulfate) or anti-CD90.2 antibody-coated Au nanorods (PSS-AuNRs)	Not reported	Intravitreal injection obscured the retinal signal and induced ocular inflammation	[57]
	Nanoscaffolds	Nanofibrous porous membrane	Diameter/porosity: 680	Bruch’s membrane thickness changes with aging, and it correlates with RPE function	[83]
Therapeutic	Gold (Au)	Nanoparticles	Diameter: 20	AuNP-labeled photoreceptor precursor transplantation provides high-resolution long-term tracking and cell survival with no toxic effects on retina or cells	[91,117]
	Core–shell-structured β-NaYF4:20%Yb, 2%Er@β-NaYF4	Nanoparticle (core–shell-structured upconversion nanoparticles (UCNPs))	Diameter: 35–40	Retinal pbUCNP injection extends the visual spectrum to the near infra-red range in mice	[15]
	Synthetic nanoscaffolds	Nanofibrous scaffolds	Diameter/porosity: 100–200	Used as a cell replacement therapy	[108]

## Data Availability

Data available in a publicly accessible repository.

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
