# Peer review of "Nano-Biomaterials for Retinal Regeneration"

_nanomaterials, 2021, doi:10.3390/nano11081880_

Round 1

Reviewer 1 Report

I think the current revised version seems OK for me

Author Response

Thank you.

Reviewer 2 Report

Review of the manuscript “Nano-biomaterials for retinal Regeneration” by Sharma et al. The manuscript reviews the state-of-the-art research activities on the use of various nanomaterials for retinal regeneration applications. The review is based on 138 scientific papers listed in the reference list. Overall, it is an interesting and readable review which covers the important recent works and directions in the field. However, the text contains many errors in the use of the English language (mainly concerning the use of articles, commas, verb noun relations) and should be thoroughly copy-edited before publication.

Some more specific comments:

Subtitle should read: A review on nano biomaterials for retinal regeneration

- Introduction: the topic “tissue regeneration” is mentioned only briefly. Why? The topic “drug delivery” is discussed in more detail. Although, the main topic is retinal regeneration.

-      Conclusion: The safety concern for human use which is still the biggest and partially unsolved challenge. This is mentioned in the conclusion, but should be made clearer. In the conclusion, it is aso implied that noninvasive nanostructure delivery will be available in the future. Is that right? This should be explained more clearly.

What about the

  • Fabrication methods of the nano biomaterials should be briefly mentioned in such a review.
  • Figure 2 is not clear to me. What is the black circle? What does the filled green circle in the middle symbolizes? Is it a structure that is formed into NP, nano wires or hybrid NPs? The big arrow indicates the translation to applications. But it is not clear where it starts. The figure should be changed and made clearer.
  • Table 1: The meaning of the “size range” should be made clearer for nano wires (length?) and nanoparticles (the smallest diameter?)

Reviewer 3 Report

This review presents insights into various emerging nanomaterials for ocular tissue engineering and regeneration. This topic is very interesting but very complicated, since there are many scales and many scientific fields. On overall point of view, this review appears to be only focused on mice. To improve this paper, following points need to be carefully addressed before acceptance.

  1. Most of the progress reviewed in this work were the research results of the past ten years. Timely insights containing more recent advances about the development in past three or five years should be provided in this work.
  2. The future improvements of nanomaterials should be deeply investigated to provide comprehensive information for retinal regeneration.
  3. For the noble metal materials, current literatures on gold (Small Methods, 2020, 4, 1900469; Advanced Science 2020, 7, 1903730), silver (VIEW 2020, 1:20200014; Angew. Chem. Int. Ed 2020, 59, 1703), and alloys (Advanced Materials 2020, 32, 2000906; Matter 2019, 1, 1669-1680) should be included.
  4. The review only reported the mice model. Other animal models using nano biomaterials should be included.
  5. Details of nano-scaffolds and its morphologies for retinal regeneration should be included in Table 1.

Round 2

Reviewer 3 Report

The paper can be accepted.

This manuscript is a resubmission of an earlier submission. The following is a list of the peer review reports and author responses from that submission.

Round 1

Reviewer 1 Report

The present review is written to present a comprehensive insight into various emerging nanomaterials such as nanoparticles, nanowires, and hybrid nanostructures that have been useful in mice for ocular tissue engineering and regeneration. Furthermore, the current status, future perspectives, and challenges of nanotechnology in tracking cells or nanostructures in the eye and its use in modified regenerative ophthalmology mechanisms have also been proposed and discussed in detail. I agree that the manuscript can be published after fully addressing several issues as following:

  1. The pictures in the review are quite limited, and the third picture has similar information to the first picture, so I think the major concern of this paper is to involve more figures in the main text.
  2. In 3.3 part “Therapeutic studies on nano-biomaterials implantation and imaging” ,the current progressive information is quite limited. It is recommended to review more recent advances and news in related fields;
  3. The format of references is not uniform and needs to be carefully rechecked: for example, references [10],[15]and many others.
  4. The authors could add the following references which would again increase the interest to general functional nanomaterial readers: Chinese Journal of Polymer Science, 2020, 38, 1149-1156; Nanoscale, 2020, 12, 11395-11415; Materials Horizons, 2020‚7, 746-761.

Reviewer 2 Report

Overall a poor review.

Although it contains 99 references the descriptions (text) are very limited.

All nano-entities are not reviewed. For example nanofibrous meshes can also play a part.

Authors have not methodically searched WoS, key paper contents have not been stidied, see:

http://website60s.com/upload/files/nanotechnology-in-regenerative-ophthalmolo_2019_advanced-drug-delivery-revie.pdf